# Electricity Generation Forecast of Shanghai Municipal Solid Waste Based on Bidirectional Long Short-Term Memory Model

**DOI:** 10.3390/ijerph19116616

**Published:** 2022-05-28

**Authors:** Bingchun Liu, Ningbo Zhang, Lingli Wang, Xinming Zhang

**Affiliations:** 1School of Management, Tianjin University of Technology, Tianjin 300384, China; tjutlbc@tjut.edu.cn (B.L.); znb211572247@163.com (N.Z.); 2School of Environmental Science and Engineering, Tianjin University, Tianjin 300072, China; 3School of Management Science and Real Estate, Chongqing University, Chongqing 400044, China; rics1103@163.com

**Keywords:** MSW generation volume forecasting, electric power generation, waste to energy, BiLSTM

## Abstract

The accurate prediction of Municipal Solid Waste (MSW) electricity generation is very important for the fine management of a city. This paper selects Shanghai as the research object, through the construction of a Bidirectional Long Short-Term Memory (BiLSTM) model, and chooses six influencing factors of MSW generation as the input indicators, to realize the effective prediction of MSW generation. Then, this study obtains the MSW electricity generation capacity in Shanghai by using the aforementioned prediction results and the calculation formula of theMSW electricity generation. The experimental results show that, firstly, the mean absolute error (MAE), mean absolute percentage error (MAPE), and root mean square error (RMSE) values of the BiLSTM model are 42.31, 7.390, and 63.32. Second, it is estimated that by 2025, the maximum and minimum production of MSW in Shanghai will be 17.35 million tons and 8.82 million tons under the three scenarios. Third, it is predicted that in 2025, the maximum and minimum electricity generation of Shanghai MSW under the three scenarios will be 512.752 GWh/y and 260.668 GWh/y. Finally, this paper can be used as a scientific information source for environmental sustainability decision-making for domestic MSW electricity generation technology.

## 1. Introduction

With the growth of the world’s population, economic development and the acceleration of urbanization, Municipal Solid Waste (MSW) has increasingly become a focus of attention [1]. At present, the large amount of MSW and improper disposal are the most challenging environmental problems faced by all countries in the world, and it is also one of the important problems of fine urban management in China. From the source, MSW is collected from household, industrial, commercial buildings and urban sources, and its generation is closely related to human activities [2]. From the perspective of MSW treatment methods, they are landfill, incineration, and composting respectively. In the volume of MSW collected, landfill accounted for 52%, incineration accounted for 45%, and composting accounted for only 3% [3]. It is urgent for China to accurately predict the amount of MSW generated, and then accurately predict the amount of electricity generated by MSW [4]. MSW generating electricity is a form of waste-to-energy (WtE); it can help bridge the gap between sustainable environment and energy supply, reduce the amount of waste sent to landfills, and generate useful electricity at the same time to achieve sustainable urban development.

In the face of the growing MSW generated by complex challenges, China has invested a lot of energy in MSW management and technological development, both to conform to national policy and try to reduce the solid waste landfill area, seeking the right MSW disposal methods, so the MSW electricity generation in our country, MSW management, plays a more and more important role, and accurate prediction of MSW generation can meet part of the energy demand and ensure effective MSW management to overcome environmental pollution. At present, the research methods used in MSW electricity generation prediction are mainly divided into the traditional statistical prediction method [5], time series prediction method [6], and combination prediction method [7]. The traditional statistical forecasting method is based on MSW and uses the quality of combustible waste in MSW to calculate and forecast the electricity generation of MSW [8]. As the waste composition index is not fixed, the prediction range of MSW electricity generation is unstable and the precision is not high. The time series prediction method effectively solves the complexity of statistical methods and has been successfully applied to a variety of complex nonlinear organic solid waste problems; the prediction accuracy is higher than that of traditional statistical prediction methods [9]. Although the single model has made a breakthrough in prediction accuracy, it still cannot reach a satisfactory height. On that basis, Miyuru Kannangara et al. [10] used the combination model of decision tree and neural network to predict the electricity generation of Canadian MSW and found that the combined prediction method had better prediction performance compared with the single prediction method. At present, it is urgent to optimize the research basis to predict the electricity generation of MSW. In recent research, the deep learning method has been gradually applied to the prediction process of MSW electricity generation, but the prediction accuracy of the simple Long and Short -Term Memory (LSTM) neural network model is often lower than that of the combined model prediction method.

In this study, a Bidirectional Long Short- Term Memory (BiLSTM) was established to predict the MSW generation in Shanghai from 2020 to 2025 by using six key influencing factors to reduce the uncertainty in the prediction model. Secondly, based on the predicted results of MSW generation, this study predicts the electricity generation of MSW and puts forward appropriate suggestions for sustainable environmental development. The contributions of this study mainly include three aspects: (1) A combined prediction model of MSW generation based on BiLSTM was established. (2) National economic indicators include Gross Domestic Product (GDP), per capita disposable income and per capita consumption expenditure, population indicators include population density and year-end resident population, social indicators include the number of urban public transport vehicles (taken as input indicators), and the new combination model is used to improve the prediction accuracy. (3) Combined with the development of the region, taking the prediction result of MSW generation, the electricity generation of MSW is reasonably predicted, and then the optimization scheme of MSW disposal is put forward to save the urban land area and improve environmental quality.

## 2. Literature Review

### 2.1. Influencing Factors of MSW Generation

MSW generation is a complex process, mainly including MSW generation volume transportation, collection and classification, MSW incineration, MSW heat electricity generation, pollutant treatment, and other aspects. The electricity generation of MSW is mainly affected by the generation of the combustible part of MSW in solid waste. The research on the influencing factors of MSW mainly focuses on the three aspects of economy, society, and population. Among them, the economic indicators commonly used by scholars include GDP [11,12], per capita consumption expenditure, per capita disposable income [13], the consumption level of residents, total retail sales of consumer goods [14], social indicators include the number of urban public transport vehicles, green space area, road area and urban green coverage rate [15], while population indicators include population number and population density [16]. The influencing factors of MSW generation required in this study are determined by scholars’ existing research results.

In addition, some scholars have further studied the relationship between MSW generation and the main influencing factors. Lakioti et al. [17] studied the impact of social, economic, and demographic factors on a small scale (i.e., family level or urban unit) and adopted factors, including family size, education level, socioeconomic status, and income level [18], together with age, household employment, and population per household [19,20] and seasons [21]. Ogwueleka [22] showed that there was a strong positive correlation between MSW generation and household size and income level, as well as high-income groups. There was a significant difference between the household size of high-income groups and average daily per-capita household waste generation, and household consumption pattern would be affected, leading to changes in the composition and quantity of household solid waste. Khan et al. [23] evaluated the generation of household solid waste from different social and economic factors, such as family income, education profession, and number of family members. The generation rate and composition of MSW were closely related to various social and economic parameters in the community and the results showed that the middle socio-economic group produced the largest amount of solid waste. Wang et al. [24] used the ESDA method to analyze the effects of population, green coverage rate, industrial structure, and road density on the amount of MSW generation, and found that population, technology, urbanization, and green coverage rate all have inhibitory effects on MSW generation. Industrial structure, number of hospital beds per capita, and road density are the driving factors of MSW generation. Cheng et al. [25] studied the relationship between GDP, population size, education level, gas permeability, and the proportion of tertiary industry and the amount of MSW generation, and found that population growth and urbanization promoted the generation of MSW, and the increase in gas permeability reduced the emission of MSW. At the same time, the proportion of the tertiary industry is significantly positively correlated with the proportion of MSW.

### 2.2. MSW Prediction Models

In recent years, a variety of forecasting methods have been applied to the MSW generation prediction; existing research can be divided into three types, as listed in Table 1: linear regression method [26], statistical analysis methods [27], and artificial neural network [28,29]. These models do not fully consider the long-term correlation between the input samples, so the ability to improve the accuracy of the prediction model of MSW generation is very limited. Sunayana et al. [30] used nonlinear autoregressive (NAR) to predict the monthly generation in Nagpur (India) in 2023, and established a classical multiplication decomposition model of simple linear regression for time series, with a maximum error of 6.34%, overcoming the data availability problem. Li et al. [27] determined three series of important parameters through statistical analysis, sampling survey, and the Analytic Hierarchy Process: the amount of waste generated by unit consumer expenditure, the distribution of consumer expenditure to activities in unit time, and the time allocation of different resident groups to activities. Noori et al. [31] developed an improved support vector machine model combined with principal component analysis (PCA) technology to predict the weekly generation of MSW, with an r^2^ of 0.75 and MRE of 3.35%. However, the SVM model is not only a small sample prediction model but also, with the increase in training data, it will consume a lot of time and computer performance, affecting the universality of the model. Sun et al. [32] used MATLAB to build an ANN neural network model and predict the future MSW generation capacity of Bangkok. In this case, a recursive neural network is introduced to improve the accuracy of MSW electricity generation prediction.

Compared with traditional statistical methods and machine learning methods, deep learning technology solves the problem that traditional statistical methods find it difficult to deal with nonlinear data. RNN is a deep learning network, and there is a recursive link in the network structure. The relationship between samples before and after learning can be considered, especially for processing time series signals. Some scholars have studied various improvement methods for the problems of gradient explosion and gradient disappearance. The emergence of the LSTM neural network effectively solves the problems existing in previous models and has achieved considerable results in the field of MSW generation. Dongjie Niu. [33] selected an LSTM model to make a long-term prediction of MSW, considering the static and dynamic change characteristics of MSW; it was found that the LSTM model had a better prediction effect compared with ANN and SVM. At present, it is difficult for a single model to achieve a better prediction effect, but the method of multi-model fusion finds it easier to improve the accuracy of the prediction model. Therefore, in order to accurately predict the generation of MSW, this study constructed the BiLSTM combined model, optimized the input index and improved the prediction accuracy of the model.

## 3. Methodology and Study Area

BiLSTM is composed of a bidirectional recursive neural network (BI-RNN) and long and short memory (LSTM), which is a standard neural network. In this case, BiLSTM is generated to solve the problems of gradient explosion and information deformation. Besides, it can effectively handle the sequential data composed of reason-time instances.

BiLSTM model can not only analyze the subtle relationship characteristics among the original index factors, but also combine the self-learning and fault-tolerant ability of the neural network, which can not only improve the prediction accuracy of MSW production amount, but also improve the network learning efficiency, and then reasonably and effectively predict the MSW power generation. The steps for predicting MSW power generation based on the BiLSTM model are shown in Figure 1.

### 3.1. Basic Principles of BiLSTM

In order to improve the learning ability of the traditional LSTM model, the bidirectional relationship of input data in the time structure is considered, rather than the single direction through the LSTM gate of input processing, and the bidirectional LSTM model fully considers the next information when processing the current time series data. This kind of two-way processing obtains more structural information through the gate mechanism and enhances the method of information intelligence. The BiLSTM model encodes the information in order to obtain the information characteristics of the data before and after, thus improving the generalization ability. The LSTM unit starts from the input sequence, and the inverse form of the input sequence has been integrated into the LSTM network. The BiLSTM model generated by the forward $h_{t}$ and backward layers $h_{t}^{\prime }$ is shown in Figure 2. Calculate forward from time 1 to time t in the forward layer to get and save the output of forwarding at each time. Calculate Backward from time t to Time 1 in the backward layer to get and save the output of the backward layer at every moment. Finally, the final output can be obtained at each moment by combining the out put results at the corresponding moments of the forward layer and the backward layer. The mathematical expression is shown in Equations (1)–(3):
(1)$$h_{t}=f\left(w_{1}x_{t}+w_{2}h_{t-1}\right)$$
(2)$$h_{t}^{\prime }=f\left(w_{3}x_{t}+w_{5}h_{t-1}\right)$$
(3)$$\sigma_{t}=g\left(w_{4}h_{t}+w_{6}h_{t}^{\prime }\right)$$
where, $W_{i\;}\left(i=1,2,\cdots ,\right)$ are six independent weight matrices, as shown below: Input the forward and backward hidden layer weights (*w*_1_,*w*_3_), hidden layer to hidden layer weight (*w*_2_,*w*_5_), hidden layer forward and backward output layer weight (*w*_3_,*w*_6_). These six weights are repeated at each time step. $\sigma_{t}$ is the final output value obtained by combining the output of the forward and backward layers.

### 3.2. Study Area

Shanghai is China’s leading economic city and financial center and aims to become the world’s top city and science and technology center. With the sustained, stable, and rapid development of the national economy, the demand for resources and the output of MSW are increasing day by day, which restricts the sustainable development of cities. How to properly solve the collection, transportation, and treatment of MSW has become the primary work of Shanghai environmental protection department. In 2019, the amount of MSW in Shanghai rose to 10.38 million tons, half of which is burned, and the Shanghai municipal government has been trying to develop a mature urban circulation system, set up the big four garbage classification system, and the city government also issued the first Chinese city waste management regulations, in accordance with those established from the source separation, the final disposal of the entire collection, and recycling chain, and the regulations took effect on 1 July 2019; it is expected that Shanghai’s recyclable waste will be better managed. It is also important to assess the recycling capacity of recyclable waste so that a recycling system can be optimized.

### 3.3. Data Source

Based on scholars’s research on the factors influencing the volume of MSW and the availability of indicator data in Shanghai, this study collected and integrated the available indicator data sets for 1978–2018 for 6 influencing factors in 3 major categories: social, economic, and demographic. Namely, GDP (CNY billion), per capita disposable income (CNY), per capita consumption expenditure (CNY), public transport vehicles numbers (car), the permanent resident population (10,000 people), permanent resident population density (people/square kilometer), for total of 246 annual data points. The data are all from the 1978–2018 Shanghai Statistical Yearbook published by the Shanghai Municipal Bureau of Statistics. Table 2 shows the details of the six indicators.

### 3.4. Prediction Model Evaluation Index

Mean absolute percentage error (MAPE), root mean square error (RMSE), and mean absolute error (MAE) were used to evaluate the prediction performance and fitting degree of the model constructed in this paper. MAPE, RMSE, and MAE are used to measure the difference between the simulated data and the model data, and also the value range. When the predicted value is in complete agreement with the real value, it is equal to 0. The greater the error, the greater the value $\left[0,+\infty \right)$.

The calculation formula of average absolute percentage error is shown in Equation (4).
(4)$$\mathrm{MAPE}=\frac{100\%}{n}\sum_{i=1}^{n}\left|\frac{{\hat{y}}_{i}-y_{i}}{y_{i}}\right|$$

The root mean square error calculation formula is shown in Equation (5).
(5)$$\mathrm{RMSE}=\sqrt{\frac{1}{n}\sum_{i=1}^{n}{\left({\hat{y}}_{i}-y_{i}\right)}^{2}}$$

The calculation formula of average absolute error is shown in Equation (6).
(6)$$\mathrm{MAE}=\frac{1}{n}\sum_{i=1}^{n}\left|{\hat{y}}_{i}-y_{i}\right|$$
where, $\hat{y}=\left\{{\hat{y}}_{1}\right.,{\hat{y}}_{2},\cdots ,\left.{\hat{y}}_{n}\right\}$ is the predicted value, $y=\left\{y_{1}\right.,y_{2},\cdots ,\left.y_{n}\right\}$ is the true value, and n is the number of indicator variables.

### 3.5. Electricity Generation of MSW Estimation in This Study

The incineration of combustible components of refuse releases a lot of heat energy. The captured heat can be used to generate steam in a boiler, which drives a steam turbine to generate electricity. Thus, the useful heat produced in a steam turbine is generated by mass combustion of usable waste parts, *M_W_*_1_ (or—organic matter), *M_W_*_2_ (paper), *M_W_*_3_ (plastic), *M_W_*_6_ (rubber and textiles), and *M_W_*_7_ (wood) can produce electricity per year *E_P_*_(*INC*, *M*-*B*)_, which can be calculated as follows (Ayodele et al., 2017):(7)$$Ep(INC,M-B)=\frac{(LHV_{W}\cdot M_{w})\times \eta }{3.6}$$
where $LHV_{W}\cdot M_{w}$ is the dot product of two vectors: $LHV_{w}=$[*LHV_W_*_1_, *LHV_W_*_2_, *LHV_W_*_3_, *LHV_W_*_6_, *LHV_W_*_7_] and $M_{w}=\left[MW1,\;MW2,\;MW3,\;MW6,MW7\right]$. $LHV_{w}\left(\mathrm{MJ}/\mathrm{KG}\right)$ is the waste LHV. The $LHV\left(\mathrm{low}\;\mathrm{humidity}\;\mathrm{and}\;\mathrm{high}\;LHV\right)$ of biologically dried combustible is shown in Table 3 [35], where $LHV_{2}\left(\mathrm{MJ}/\mathrm{KG}\right)$ is the new energy content of biologically dried MSW components. The individual available component $M_{w}$ can be obtained by using the following methods:
(8)$$M_{W(C)}=\frac{\sum {}_{t=1}^{n}M_{W(C)}(t)}{n}$$
where (*C*) refers to 1, 2, 3, 6, and 7, respectively, to individual waste components: organic matter, paper, plastics, rubber, textiles, and wood. Annual electric energy *E_P_*_(*INC*, *RDF*)_ is used through the burning of *RDF*.
(9)$$E_{P(INC,RDF)}=\frac{LHV_{RDF}\times M_{U(INC)}\times \eta }{3.6}$$

The *LHV_RDF_* is 17.9 MJ/kg [35]. The conversion efficiency of mass incineration is *η* = 0.29 [36] and 0.26 *RDF* incineration [37]. *M_U_*_(*INC*)_(tons/year) is the average mass of waste burned each year by
(10)$$M_{U(INC)}=\frac{\sum {}_{t=1}^{n}M_{U(INC)}(t)}{n}$$

*M_U_*_(*INC*)_(*t*) is the amount of waste (*n*) (in tons) available for incineration during the project period. For the quality of waste composition under different scenarios, see Figure 3.

## 4. Results

### 4.1. Model Accuracy

In order to effectively evaluate the performance of BiLSTM in the prediction of MSW, this paper adopts traditional machine learning and deep learning prediction methods as a comparative experiment. Based on the same input time series, the learning of each model is tested and its errors are compared and analyzed. In the experiment, support vector regression (SVR), gate regression unit (GRU), bidirectional and support vector regression (BI-SVR), and bidirectional and gate regression (BI-GRU) are used to predict the time series, and comparative tests are carried out. Besides, the proportion of the training set to the test set was the same as that of the BiLSTM model, and five comparative experiments were conducted. In order to objectively evaluate and describe the performance of these six prediction models, the prediction error values of each model are calculated according to the above formulas. The BiLSTM neural network contains four parameters that affect the prediction accuracy of the model, including the learning rate, the time step of each layer, the number of Hidden_layers of each layer and the number of training epochs. When the number of Hidden_layers gradually increases, the number of Hidden_layer neurons has little effect on the results, and the prediction error curve is relatively stable. In the training process of the model, the setting of a single parameter is different, but other parameters are the same, so as to find the best prediction model. Each parameter setting in the proposed model is shown in Table 4. The process steps of adjusting parameters are as follows: (1) Adjust the BiLSTM model architecture suitable for time series prediction and test the applicability of the model architecture to the data set. (2) The optimal number of BiLSTM Hidden_layers is determined according to the size of the data set. (3) According to the data characteristics of the data set, the optimal activation function is selected from the four types of sigmoid, tanh, relu, and linear functions. (4) The learning rate value, time step, and batch size are determined by grid search. At the beginning of the experiment, the default super parameter setting is used to observe the change in loss, preliminarily determine the range of each super parameter, and then adjust the parameters. For each super parameter, we only adjust one parameter each time, and then observe the loss change until the optimal parameter is determined.

The MAE, MAPE, and RMSE experimental results of the original test set are shown in Table 5. Among the six prediction model algorithms, LSTM has the largest prediction error, and the traditional SVR algorithm is second only to the LSTM algorithm. The values of MAE, RMSE, and MAPE of LSTM and SVR are 163.23, 19.42, 176.32, and 163.28, 19.32, 183.24, respectively. Compared with the traditional SVR algorithm, the MAPE value of the BI-SVR prediction algorithm is significantly decreased, and the prediction accuracy is significantly improved, among which the MAPE value of the BI-SVR algorithm and SVR algorithm are 14.32 and 19.32, respectively. Compared with BI-GRU and BI-SVR, the GRU and SVR algorithms show little change, but they also play a role in improving the prediction performance of the model. Compared with other algorithms, the prediction error of the BiLSTM combined model is the smallest, and the MAPE value is 7.390.

### 4.2. Predicted MSW Generation

#### 4.2.1. Scenarios Setting

The study sets different scenarios and calculates the value of each index according to different scenarios, then the index data sequence under different scenarios is input into the prediction model to reduce the uncertainty of model prediction. Combined with the historical data of six indicators and the planning of macro-economic, consumption, and population indicators, the characteristics and trends of each indicator are reasonably analyzed, so as to effectively predict the amount of MSW generation. In this study, three scenarios were established based on different economic development conditions to predict the MSW output in Shanghai from 2020 to 2025. Scenario 1 is the low-growth scenario, which will continue to maintain the recent development trend of the country and reasonably calculate the lowest non-negative growth rate of social, economic, and demographic indicators, according to the historical growth rate of data series, to predict the amount of MSW generation in Shanghai. Scenario 2 is the base growth scenario, which is based on the average year-on-year growth rate of each indicator from 1978 to 2018, which is more consistent with the development trend of each indicator feature from 2019 to 2023, so will be more accurate. Scenario 3 is the high-growth scenario, which maintains a high-growth-rate change trend, according to the changes in historical data series. Based on the data characteristics from 1978 to 2018, this study calculates the average year-on-year growth rate of each indicator and increases 1.2-times based on the basis of the average growth rate. The growth rates of data series under different scenarios are shown in Table 6.

#### 4.2.2. Forecast Results of MSW Prediction

By comparing the error values of the single prediction model and combined prediction model, this study found that the BiLSTM model has the highest prediction accuracy, and used the BiLSTM combined model to predict the MSW generation amount of Shanghai in six years from 2020 to 2025. The index characteristics under different scenarios are input into the optimal prediction model to reasonably predict the generation amount of MSW in Shanghai. The prediction results of MSW generation under different situations are shown in Figure 4. The prediction result of Scenario 1 shows that the generation of MSW will decrease slowly in the future, from 10.38 million tons in 2019 to 8.82 million tons in 2025. In Scenario 2, the variation trend of MSW generation in Shanghai is relatively gentle, increasing from 10.38 million tons in 2019 to 11.36 million tons in 2023, and then sees an inflection point, increasing to 12.84 million tons in 2025. Scenario 3 is dominated by an overall upward trend, rising from 10.38 million tons in 2019 to 17.35 million tons in 2025. In conclusion, the amount of MSW produced in Shanghai will fluctuate from 8.82 million tons to 17.35 million tons in 2025.

Under the baseline growth and high-growth-rate scenarios, although there will be an inflection point in 2023, the amount of MSW will show an overall upward trend. In order to solve the MSW to speed up growth, promoting economic and social development, in the “difference”, a new stage of development, taking the thought of socialism with Chinese characteristics in Xi Jinping’s new era as a guide, the party’s 19th session of 2, 3, plenary meeting spirit should be fully implemented: accelerate the MSW classification on the classification, classified collection, classification, transport, construction of facilities for treating, filling up the treatment capacity gap, improve the urban environmental infrastructure, improve the ecological environment, upgrade the modernization of treatment capacity, promote the formation of an MSW classification and treatment system compatible with economic and social development and comprehensively promote the construction of an incineration treatment capacity. In areas where the daily garbage collection volume exceeds 300 tons, it is necessary to speed up the development of garbage treatment, mainly through incineration, build appropriately advanced incineration treatment facilities, commensurate with the daily garbage collection volume, and basically achieve “zero landfill” of native domestic garbage by 2023.

### 4.3. Electrical Energy Generation Potential of MSW

During the 14th Five-Year Plan period, Shanghai will continue to improve the ability of waste incineration capacity, which will become an important method for the disposal of MSW in the future. Therefore, it is very important to reasonably predict the electricity generation of MSW and improve the disposal efficiency of MSW. Based on the BiLSTM combined model, this study predicts the generation amount of MSW using different scenarios and predicts the electricity generated by the disposal of solid waste by using the garbage electricity generation calculation formula. The estimated results are shown in Figure 5.

Under the low-growth-rate scenario, Shanghai’s MSW electricity generation shows a downward trend, and its electricity generation decreases from 303.22 GWh in 2020 to 260.67 GWh in 2025. Under the baseline growth rate scenario, the electricity generation of Shanghai MSW remains stable, and the electricity generation of Shanghai MSW fluctuates between 335.73 GWh and 379.47 GWh. Under the high-growth-rate scenario, the electricity generation of Shanghai MSW keeps rising and is expected to decrease by the end of 2023. However, overall, the electricity generation in Shanghai MSW keeps increasing and is expected to reach 512.75 GWh in 2025. These results indicate that MSW electricity generation has great energy potential.

## 5. Discussion

### 5.1. Analysis of Driving Factors of MSW Generation

The factors affecting the generation of MSW in Shanghai are complex; Shanghai MSW generation shows a strong growth trend, resulting in the continuous growth of MSW electricity generation potential. MSW electricity generation potential is generated by the mass combustion of combustible components of available MSW, so the amount of MSW generation is a decisive factor in MSW electricity generation. According to the predicted generation of the aforementioned MSW and the electrical energy generation potential of MSW, this study analyzed the reasons for the driving factors and put forward suggestions for sustainable MSW management. The sustainable growth of MSW generation in Shanghai may be related to the following aspects. First, China’s economy has sustained and rapid growth. At present, China’s economic development model from high-speed development to high-quality development, for the growth of MSW, has laid a foundation. Second, the increase in MSW is related to the rapid growth of the population. With the continuous development of cities, the population number is increasing. The increase in population will inevitably lead to more generation activities and consumption materials, resulting in a large amount of MSW. Third, the improvement and expansion of urban construction level and scale, the continuous improvement in garbage collection and transportation, leading to the rapid growth of MSW, results in an urgent need for the government to rationally plan and utilize the value of MSW. Fourth, the improvement of Shanghai residents’ living standards has promoted the increase in per-capita disposable income and expenditure. The per capita consumption expenditure of Shanghai MSW increased from 488 CNY in 1978 to 46,015 CNY in 2018, and the per-capita disposable income of Shanghai MSW increased from 560 CNY in 1978 to 60,231 CNY in 2018. People’s purchasing power has increased. In this way, the quantity and types of consumption will increase, and more MSW will be generated.

### 5.2. Suggestions for Sustainable MSW Management

The improper disposal of MSW affects public health and harms the people living in the surrounding areas. Landfill will cause smell, dust, wind-blown garbage, visual interference, noise, and traffic congestion to varying degrees. In addition, it will also cause land resource dissipation and pollution. The antibiotics in landfills leaching to nearby environment by leachate may threaten ecosystem health, leading to a high concentration of pollutants that are difficult to treat [38]. Due to greenhouse gas emissions, leachate and overpopulation city land usability problems do not conform to the requirements of the sustainable development of the environment, and we are seriously in need of a shift from the traditional methods of landfill waste electricity generation. Compared with landfill, waste into energy, or MSW incineration, electricity generation is a good way to replace fossil fuel combustion. The development of this technology in China’s MSW treatment and resource utilization has been greatly improved and can be used as a sustainable alternative method. The utilization of poultry waste for energy generation is feasible and environmentally benign. Arshad M et al. [39] estimated the waste generated from poultry farming and discussed technology for the conversion of poultry waste into biogas. Generating electricity from poultry waste is feasible and environmentally sound to reduce the reliance on fossil fuels.

Global economic development has led to increased demand for energy, and the energy supply chain is overburdened. Fossil fuel reserves are developed to meet high energy demand, and their combustion is becoming a major source of environmental pollution. There is an urgent need to find safe, renewable, and sustainable sources of energy. WtE can be considered as an alternative energy source that is economically and environmentally sustainable. MSW electricity generation is considered as a kind of renewable energy, and WtE is a win-win strategy that can eliminate waste and generate energy, issues that has caused widespread concern around the world [40]. MSW is a major contributor to the development of renewable energy and sustainable environment. These data for MSW electricity generation under the three scenarios indicate that MSW incineration can be an option and contribute a larger share in the national energy matrix. In addition, the growth of large-scale incineration for national energy will support the insertion of intermittent renewable energy, which is required to provide a stable output in national system energy. Thus, MSW can play a vital role in offsetting fossil fuel consumption and increasing the share of renewable energy.

MSW electricity generation is one of the main methods of biomass electricity generation. On 22 October 2016, the Ministry of Housing and the National Development and Reform Commission, as well as other departments, jointly issued opinions on further strengthening the incineration treatment of urban household garbage, which first affirmed the role of household garbage incineration treatment, and put forward the “planning first”, speeding up construction, to make up the shortcomings of MSW treatment, take the construction of waste incineration treatment facilities as the key to maintain public security, promoting the construction of ecological civilization, improve government governance capacity and strengthen urban planning, construction, and management, which shows the determination of the state to firmly support waste treatment to adopt incineration electricity generation. In December 2016, the National Energy Administration issued the “13th Five-Year Plan for biomass Energy Development” in the “Development layout and Construction key points”, which aims to: “Encourage the construction of waste incineration cogeneration projects. Accelerate the application of modern waste incineration treatment and pollution prevention and control technologies to improve the environmental protection of waste incineration electricity generation. Strengthen publicity and public opinion guidance to avoid and reduce the nimby effect.” This further reflects the country’s high attention to biomass energy. In September 2020, the National Development and Reform Commission issued the “Implementation Plan for Improving the Construction and Operation of Biomass Electricity Generation Projects” to clarify the subsidy methods for new projects in the transitional period. In October 2020, the Ministry of Finance and other departments issued the Supplementary Circular on Subsidy Funds to the Opinions on Promoting the Healthy Development of Non-Water Renewable Energy Electricity Generation. The state provides practical financial support for renewable energy generation. A series of national policies will boost the generation of MSW in the future.

The development of existing MSW electricity generation technology will be the main trend of MSW treatment in China in the next few years, and incineration plants are likely to further develop into mainstream applications. Because of the diversity of MSW, how to effectively convert MSW into energy is the main challenge faced at present. In terms of controlling the generation of MSW, the government must strive to standardize the classification and collection of MSW nationwide, so as to achieve a balance between urbanization and waste flow. For example, we can learn from the practice of developed countries and set up a garbage collection system. There is a huge amount of MSW in China. Therefore, improving the treatment and recycling efficiency of MSW is of great positive significance to environmental protection, resource conservation, economic development, and human health protection. The environmental protection measures of MSW treatment in order to popularize the electricity generation technology of solid waste, must be further discussed. The ideal MSW treatment, technology should be a cost-effective system that promotes recycling, reduces emissions, and solves MSW treatment problems in a sustainable manner. Prudent policies should, therefore, be adopted to strengthen prevention, reuse, and recycling, while promoting waste-generating electricity generation.

## 6. Conclusions

The improper disposal of MSW, as a significant environmental problem, restricts regional and national economic development and people’s quality of life. Accurate prediction of MSW production and reasonable estimation of MSW electricity generation can help the environmental sanitation administrative departments to plan the scale of household waste disposal facilities and land use and avoid the waste of land resources. This study first collected six indicator variables, then indicators of GDP, the number of public transport operating vehicles, the per-capita consumption expenditure, per-capita disposable income, population density, and the population of permanent residents as input variables, input into the BiLSTM combination model to forecast the Shanghai city life garbage output, and finally, with the help of an urban living garbage electricity calculation formula, a reasonable estimate for Shanghai’s city life garbage output was established through the experiment, leading to the following conclusions:
(1)The electricity generation capacity of MSW is related to the gross regional product, the number of public transport operating vehicles, per-capita consumption expenditure, per-capita disposable income, population density, and the permanent resident population at the end of the year, which can be used as the input variable of the model to effectively predict the electricity generation capacity of MSW.(2)In this study, the BiLSTM combined model was selected to predict the MSW generation in Shanghai. The experimental results show that the MAPE of the combined prediction model is 7.390. Compared with machine learning and a single prediction model, this model can predict the MSW generation in Shanghai more accurately.(3)With the help of the calculation formula of MSW electricity generation and combined with the predicted amount of MSW generation above, a reasonable electricity generation estimate for Shanghai MSW can be obtained. Based on the changes in MSW production and electricity generation obtained from the research, exploring new technologies and maximizing the utilization of MSW will be the main goals in the future.

## Figures and Tables

**Figure 1 ijerph-19-06616-f001:**
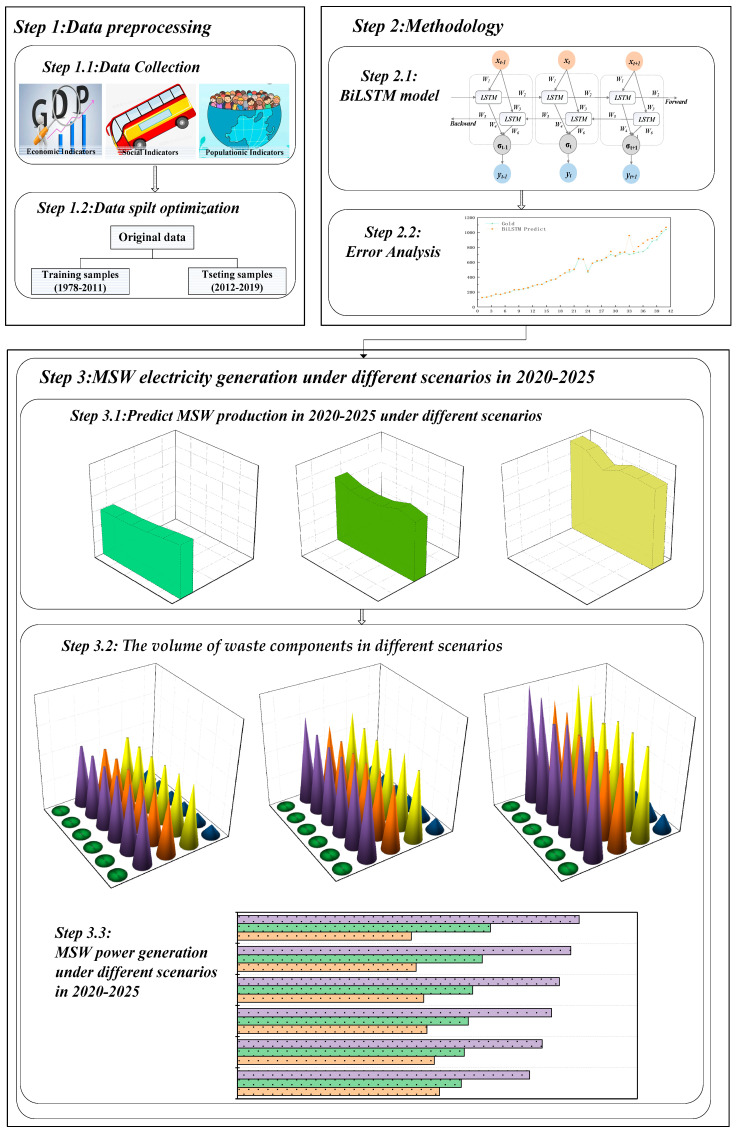
Logical structure diagram of this research.

**Figure 2 ijerph-19-06616-f002:**
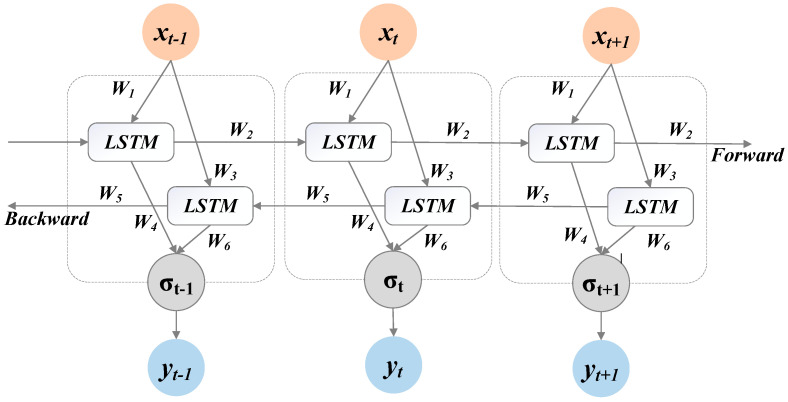
Structure diagram of Bidirectional Long Short-Term Memory (BiLSTM).

**Figure 3 ijerph-19-06616-f003:**
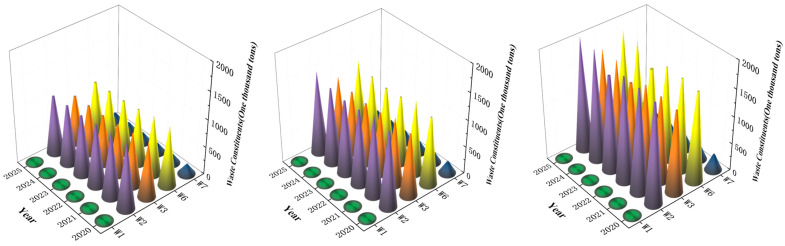
Volume of waste constituents under different scenarios.

**Figure 4 ijerph-19-06616-f004:**
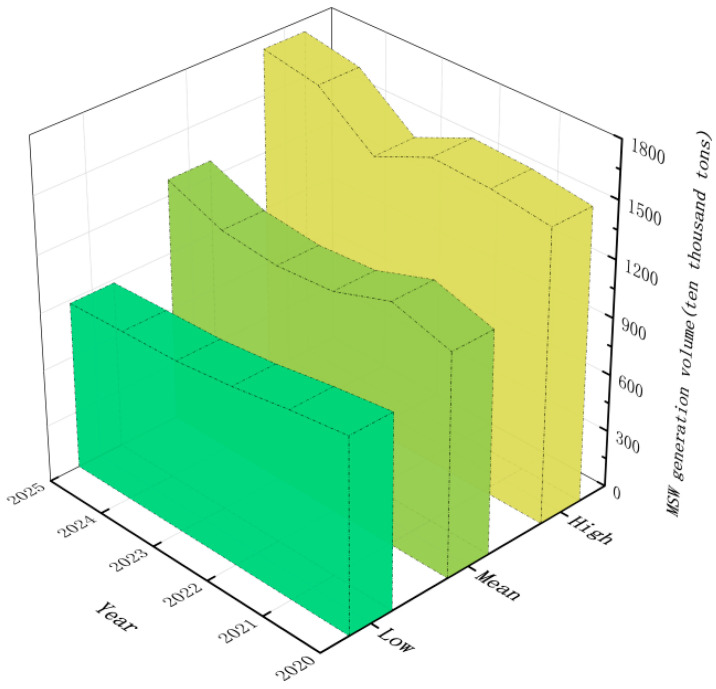
Scenario prediction of MSW generation in Shanghai.

**Figure 5 ijerph-19-06616-f005:**
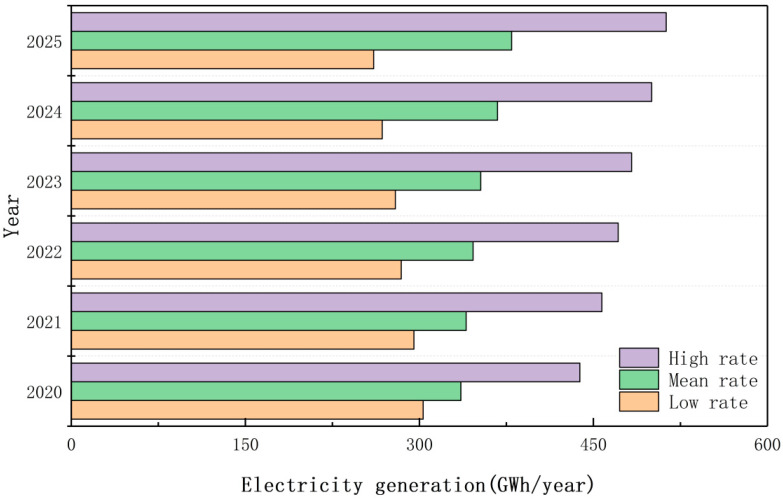
MSW electricity generation under different scenarios.

**Table 1 ijerph-19-06616-t001:** Research and comparison on prediction of Municipal Solid Waste (MSW).

Methods	Models	Authors (Year)	Cases	Performance
linear regression method	nonlinear autoregressive	Sunayana et al. (2021)	India	maximum error of 6.34%
	regression analysis	Ghinea et al. (2016)	America	-
statistical analysis methods	statistical analysis	Li et al. (2011)	Beijing	-
artificial neural network	Artificial neural network	Alidoust et al. (2021) [34]	-	R^2^ = 0.98
	ANN neural network	Sun et al. (2017)	Bangkok	R^2^ = 0.96
	support vector machine	Noori et al. (2009)	Mashhad	MRE: 3.35%

- This symbol indicates no mention of performance.

**Table 2 ijerph-19-06616-t002:** Six indicator statistics.

	Max	Min	Average	Std. Dev
MSW	1038	108	474.54	247.95
Gross regional product	32,679.87	272.81	8071.50	9595.54
Permanent resident population at year-end	2424	1098	1662.07	470.75
Per capita disposable income of urban households	60,231	560	16,637.90	18,126.98
Per capita consumption expenditure of urban households	46,015	488	12,127.02	12,873.64
Number of urban public transport vehicles operating	23,516	2983	13,011	6824.21
The population density	3823	1785	2670.78	720.99

**Table 3 ijerph-19-06616-t003:** Biological drying hypothesis.

Waste Constituents	Class	Moisture Content (% wb)	Water Reduction Via Biodrying (%)	Organic Matter Reduction Via Biodrying (%)	LHV1 (MJ/kg)	LHV2 (MJ/kg)
Organics	W1	84.8	75	16	4.4	11.3
Paper	W2	12.2	60	8	11.7	13.2
Plastics	W3	14.8	35	0	37.7	38.1
Glass and ceramics	W4	2.4	0	0	0.0	0.0
Metal	W5	2.7	0	0	0.0	0.0
Textiles and Rubber	W6	7.8	60	6	17.2	21.0
Wood and others	W7	5.4	45	6	9.8	12.3

**Table 4 ijerph-19-06616-t004:** Parameter setting for the BiLSTM neural network.

Model	Time Step	Learn Rate	Batch_Size	Hidden_Layer	Epoch	Mape (%)
BiLSTM	2	0.01	2	32	5000	11.236
	2	0.01	2	64	10,000	9.626
	2	0.001	2	64	10,000	7.390
	2	0.001	3	64	10,000	10.428
	2	0.001	3	128	10,000	12.528

**Table 5 ijerph-19-06616-t005:** Comparison of prediction performances using deep learning models.

Model	MAE	Mape (%)	RMSE
SVR	163.28	19.32	183.24
GRU	173.82	17.32	163.23
LSTM	163.23	19.42	176.32
Bi-SVR	128.32	14.32	132.73
Bi-GRU	123.53	18.32	125.53
BiLSTM	42.31	7.390	63.32

**Table 6 ijerph-19-06616-t006:** Growth rate of each indicator under different situations.

Scene Category	The Population Density	Number of Urban Public Transport Vehicles in Operation	Permanent Resident Population at Year-End	Gross Regional Product	Per Capita Disposable Income	Per Capita Consumption Expenditure
Scenario 1	0.0014	0.0125	0.0081	0.0137	0.0173	0.0334
Scenario 2	0.0020	0.0593	0.0280	0.0230	0.0210	0.0480
Scenario 3	0.0121	0.0745	0.0254	0.0353	0.0346	0.0545

## Data Availability

The data used to support the findings of this study are available from the corresponding author upon request.

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
