# Peer review of "Electricity Generation Forecast of Shanghai Municipal Solid Waste Based on Bidirectional Long Short-Term Memory Model"

_ijerph, 2022, doi:10.3390/ijerph19116616_

Round 1

Reviewer 1 Report

I revised the work entitled: “Electricity Generation Forecast of Shanghai Municipal Solid 2 Waste Based on bidirectional long and short time memory 3 Model” and concluded that, in the current version, there are numerous critical problems to make possible its consideration for publication in this Journal.

This work consists of a case study but it should introduce some scientific novelty anyway and, in these pages, I could not find it.

The methodology and the model used for this research, should be better described and more details are needed.

Considering that the research is based on a model, the state of the art the description of different approaches used in the same field, should be widely more extended than what present in this work. Moreover, details about the provenience of data used, their reliability and variability, would be also helpful. The MSW were treated globally; however, considering the dimensions of the city considered for this research, the composition might vary significantly and, with it, the percentage or reusable material, the LHV and other crucial parameters.

The list of reference should be widely extended, according to the previous comments.

Some diagrams are not meaningful in the present form; see for instance Figure 4, which does not provide additional information.

It seems that the text has been written very quickly and the quality of the discussion was not ensured. For instance, I really did not understand the meaning of the following sentence: “There is a direct correlation between the amount of MSW generation and the six influencing indicators, so there is an indirect correlation between the amount of MSW generation and the six influencing indicators”.

In conclusion, at this stage, I must reject the article.

Author Response

We would like to thank you for the insightful comments and suggestions, which we find very helpful in improving the paper's quality. We have duly incorporated all the critical comments and suggestions in this manuscript as best as possible. 

Reviewer 2 Report

Comments

Manuscript ID ijerph-1739984

Electricity Generation Forecast of Shanghai Municipal Solid Waste Based on bidirectional long and short time memory Model

Authors have reported a prediction of municipal Solid Waste electricity generation from Shanghai based on construction of bidirectional long and short time memory model, determines the economic, demographic and social indicators related to Municipal Solid Waste, with six influencing factors of Municipal Solid Waste generation as input indexes, using Bi-LSTM model to study the characteristic of input indexes, to realize the effective forecast of Municipal Solid Waste generation. This paper is very useful for scientific information source for environmental sustainability decision-making of domestic Municipal Solid Waste electricity generation technology.

This work is interesting, which is a significant advancement over existing knowledge, but it needs substantial improvements before considering for publication. The publication is recommended, subjected to revision as mentioned below in comments to the authors.

Specific comments for authors

# Abstract should contain some quantitative information also.

# There are several typographical and grammatical mistakes which should be corrected.

# Aims and object, please clearly indicate the main points undertaken in aims and objective section

# the abbreviation used must be explained on their first appearance, or provide separate list of abbreviations

# Figure 1. Logical structure diagram of BI-LSTM prediction model. Resolution is low?

# Authors must include studies reporting the same in tabulated form. The data in tabulated form will be highly useful for the reader

# Figure 3. Quality of waste composition under different scenarios. Resolution is low?

# Discussion is poor, authors need to focus on discussion to make it more interesting

# please support your statements in discussion section with relevant references

# References are not uniform, please revise, update according to journal format

# The introduction and result and discussion part should be improved and the results should be interpreted with latest references to make it more understandable for the readers, cite more latest studies related to topic under investigation, some suggestions are as; which might be helpful for updating the information.

Arshad, M., et al., Electricity generation from biogas of poultry waste: An assessment of potential and feasibility in Pakistan. Renewable and Sustainable Energy Reviews, 2018. 81: p. 1241-1246.

Moreover, authors can update the introduction in the light of existing studies that Electricity Generation form Municipal Solid Waste is more beneficial versus other conventional energy resource. Give advantages of energy Generation form Waste versus fossil fuels and biological like biodiesel energy resource, which are pollution causing and costly. Some related references are mentioned for authors guidance as an option  

Author Response

(The authors gave the same response as above.)

Round 2

Reviewer 1 Report

The quality of the article has been clearly improved. Now the text contains more useful information, the description of results has been deepened and the quality of diagrams is improved. 

I want to reward these efforts, thus I suggest to consider the article for publication.

However, I maintain the previous doubts on the novelty of this work; under this point of view, the article is weak.

Reviewer 2 Report

The authors updated the Ms in the light of points raised, so accepted for publication in current form.

Thank you

This manuscript is a resubmission of an earlier submission. The following is a list of the peer review reports and author responses from that submission.